# Exosomes are the Driving Force in Preparing the Soil for the Metastatic Seeds: Lessons from the Prostate Cancer

**DOI:** 10.3390/cells9030564

**Published:** 2020-02-28

**Authors:** Saber H. Saber, Hamdy E. A. Ali, Rofaida Gaballa, Mohamed Gaballah, Hamed I. Ali, Mourad Zerfaoui, Zakaria Y. Abd Elmageed

**Affiliations:** 1Laboratory of Molecular Cell Biology, Department of Zoology, Faculty of Science, Assiut University, Assiut 71515, Egypt; ssaber@zewailcity.edu.eg; 2Department of Pharmaceutical Sciences, Rangel College of Pharmacy, Texas A&M Health Science Center, College Station, TX 77843, USA; haali@tamu.edu (H.E.A.A.); gaballa@tamu.edu (R.G.); gab_moh@tamu.edu (M.G.); alyismail@tamu.edu (H.I.A.); 3Department of Surgery, School of Medicine, Tulane University, New Orleans, LA 70112, USA; mzerfaoui@tulane.edu

**Keywords:** castrate resistant prostate cancer, tumor microenvironment, stromal cells, exosomal cargo

## Abstract

Exosomes are nano-membrane vesicles that various cell types secrete during physiological and pathophysiological conditions. By shuttling bioactive molecules such as nucleic acids, proteins, and lipids to target cells, exosomes serve as key regulators for multiple cellular processes, including cancer metastasis. Recently, microvesicles have emerged as a challenge in the treatment of prostate cancer (PCa), encountered either when the number of vesicles increases or when the vesicles move into circulation, potentially with an ability to induce drug resistance, angiogenesis, and metastasis. Notably, the exosomal cargo can induce the desmoplastic response of PCa-associated cells in a tumor microenvironment (TME) to promote PCa metastasis. However, the crosstalk between PCa-derived exosomes and the TME remains only partially understood. In this review, we provide new insights into the metabolic and molecular signatures of PCa-associated exosomes in reprogramming the TME, and the subsequent promotion of aggressive phenotypes of PCa cells. Elucidating the molecular mechanisms of TME reprogramming by exosomes draws more practical and universal conclusions for the development of new therapeutic interventions when considering TME in the treatment of PCa patients.

## 1. Introduction

Prostate cancer (PCa) is the most common adenocarcinoma in American and European men, after skin cancer [1,2]. As estimated by the American Cancer Society, approximately 174,650 new cases and 31,620 deaths from PCa were predicted annually in the United States as of 2019 [3]. In early-stage PCa, the cancer cells remain sensitive to androgens; therefore, androgen deprivation therapy is the most effective treatment typically offered to these PCa patients [4]. Over time, however, the cancer cells become androgen insensitive, and chemotherapy agents, such as docetaxel, are one clinical option to treat androgen-independent and metastatic castrate-resistant PCa (mCRPC), a stage at which the clinical outcomes of the PCa patient are inferior [5,6]. CRPC is characterized by progression, despite the patient living with castrate levels of testosterone < 0.5 ng/mL [7]. The mechanisms proposed to illustrate this phenomenon include androgen receptor (AR) gene mutation, AR splice variant expression, AR overexpression, an increase in the expression of the activator transcription factors, and up-regulation of the androgen synthesis enzymes, such as CYP17 [8,9,10,11,12,13]. Therefore, although castration levels of the androgen are present in CRPC, the AR signaling pathway remains active. Understanding these pathways will help in the development of new targeting agents to block the AR pathway. These targeting agents include abiraterone, which blocks CYP17A1, a microsomal enzyme involved in two critical steps of testosterone biosynthesis [14,15,16], whereas Orteronel (TAK-700) and Galeterone (TOK-001) work as AR blockers by inhibiting CYP17 [17,18,19]. Common AR antagonists include Enzalutamide (MDV 3100), ARN-509, and ODM-201, which are introduced as therapeutic agents against mCRPC [20,21]. Many of the novel cytotoxic chemotherapeutic agents developed in recent years, such as docetaxel and cabazitaxel, are associated with an increase in the overall survival of mCRPC patient from 9–18 months to > 30 months [22,23,24,25]. PCa expresses tumor-associated antigens, which make cancer cells a target for vaccines [26]. Immunotherapy is an attractive therapeutic approach for treating PCa. For example, Sipuleucel-T is a cell-based immunotherapy and PROSTVAC-VF is a recombinant vaccine that consists of two vectors encoding prostate-specific antigen (*PSA*) and three immune co-stimulatory agents [27]. Although the mCRPC treatment landscape has developed significantly in the last decade, nonetheless mCRPC patients continue to face a variety of therapeutic challenges that require additional research attention. Today, the impact of the tumor microenvironment (TME) in prostate cancer development and metastasis is commonly highlighted throughout the related literature.

## 2. The Soil/Seed Analogy: Tumor Microenvironment (TME) and Tumor Cells

Analysis of the TME has been out of reach for many decades, with studies in this area only recently gaining significant momentum in cancer research. The relation between cancer cells and their TME is quite similar to the “seeds and soil” relationship, which explains the tactical role of the TME in cancer evolution and progression as a result of the stimulatory or inhibitory signals that the TME provides [28]. The TME includes the diverse cells in the vicinity of the tumor, such as fibroblast, endothelial, immune, fat, neural, epithelial, and mesenchymal stem cells [29], as well as the soluble and insoluble factors, extracellular matrix and exosomes [30]. Although multiple studies have focused on the modulating role of soluble factors on the TME, new evidence for the potential role of exosomes in altering the TME and promoting aggressive tumor behavior has now been documented [31].

## 3. Tumor-Associated Exosomes Modulate the TME and Prepare the Metastatic Niche

### 3.1. Exosomes, Biogenesis, Trafficking, Uptake and Exosomal Cargo

Cells communicate with each other by releasing different types of extracellular vehicles (EVs), such as exosomes, which are cup-shaped bi-layered membrane nanovesicles (30–120 nm in diameter), into their local microenvironment and the circulatory system. EVs are small, double-membrane bodies released by normal and abnormal cells and are classified into three main types based on the size of vesicles. The typical size of EVs ranges from 100 nm to 1µm, exosomes from 30 to120 nm, and apoptotic bodies from 500 nm to 2 µm in diameter [32]. Exosomes are intraluminal vesicles that are derived from multivesicular bodies through a process of endosome ripening, in which the vesicles either fuse with lysosomes to degrade their cargo or fuse with the plasma membrane to release exosomes into the extracellular matrix [33,34,35]. The number and composition of exosomes depend on the physiological cellular activities in which exosomes are involved in. Exosomes can be characterized by a set of exosome markers expressed on the outer membrane of these vesicles or enclosed in their cargo. The most common protein contents of exosomes are tetraspanins (CD63, CD9, CD81, and CD82), ESCRT-I associated protein (TSG101), lysosome-associated membrane glycoproteins (LAMP-1 and 2B), MVB-associated protein (Alix-1), heat shock proteins (HSP60, 70, and 90), adhesion molecules (CD54 and CD11b), major histocompatibility molecules (MHC-I and II), Ras-related proteins (Rabs), and membrane-binding proteins (annexins) [36]. In addition to proteins, exosomes carry in their cargos RNAs (microRNAs, mRNAs, and lncRNAs), DNAs, and lipids, which have been previously reported [37].

Several mechanisms regulate exosomal trafficking and release. Kirsten Rat Sarcoma (RAS)-associated binding (Rab) proteins mediate exosomal trafficking and their release to the extracellular matrix [38]. Interestingly, cancer cells overexpress different EV-associated biogenesis machinery, such as components of the Endosomal Sorting Complex Required for Transport (ESCRT), syntenin, heparanase, YKT6, amplifying Rho/ROCK, EGFRvIII, H-RASV12, and proto-oncogene Src signaling, which subsequently causes the release of a significantly higher quantity of exosomes than normal cells would release [39,40,41,42,43,44,45,46,47,48]. Hypoxic conditions and a low-pH TME can also positively impact the release and uptake of exosomes by cancer cells [49,50,51,52]. In addition, p53 plays a significant role in enhancing exosome secretion, although its mechanism of release remains unknown [53]. Another factor is the ceramides, which can induce exosomal budding into the multivesicular endosomes and inhibit the neutral sphingomyelinase 2 (nSMase2) enzyme, which is a rate limiting enzyme in ceramide biosynthesis, and thereby suppresses exosomal release [54,55,56].

Essentially, the composition of the exosomes determines their autocrine, paracrine, and endocrine functions. The protein cargo of exosomes depends mainly on the ubiquitination process, the plasma membrane anchor tags provided by myristoylation, prenylation, or palmitoylation, and the transmembrane glycoprotein CD43 [57,58]. Even so, it should not be assumed that exosomal RNAs are randomly loaded; in contrast, the process is tightly regulated by specific shuttle events, such as SUMOylated heterogeneous nuclear ribonucleoprotein A2B1 (hnRNPA2B1), which specifically binds to microRNAs (miRNAs or miRs) that contain the ‘shuttling’ motif GGAG, thereby resulting in their upload into exosomes [59]. In addition, it has been suggested that AGO2, a protein accompanied with the RNA-induced silenced complex (RISC), regulates the loading of miRNAs into exosomes [60]. The overexpression of miRNAs and low expression of their target mRNAs are key factors for the loading of these miRNA sequences into exosomes [61]. Furthermore, exosomes protect their cargo against enzymatic degradation through trafficking in the circulatory or extracellular environment [38]. Interestingly, exosomes mirror the metabolic status of the cells of origin, which reflects the emerging role of exosomes as a fingerprint in the diagnosis of many diseases, including PCa [62,63,64,65]. As established by multi-omic studies, tumor-derived exosomes shuttle many bioactive molecules, such as proteins, nucleic acids, and lipid molecules, to and from the TME, which directly exerts phenotypic changes in recipient cells and promotes cancer progression [66]. In the same context, exosomes can communicate between two different cells through 1) transfer of bioactive molecules in their cargo to activate/suppress signaling pathways in target cells, 2) receptors shuttling between donor and recipient cells to alter cellular activities, 3) transfer of fully functional proteins to perform specific functions in target cells, and 4) providing new genetic information to recipient cells to gain new traits [67].

The clinical relevance of exosomes in cancer progression is well established, and differences have been observed in the exosomal cargo of cancerous and normal cells. As such, exosomes have a significant role in the early diagnosis of cancer [68] In PCa patients, several drawbacks remain with regard to the clinical utility of prostate specific antigen (PSA) and carbohydrate antigens as diagnostic markers [69]. However, biopsy is a decisive method of diagnosis, and novel early diagnostic biomarkers are required for clinical applications. The exosomes isolated from PCa blood, urine or saliva can be used as predictive biomarkers. It was reported that PCa patient-derived exosomes shuttled Epidermal Growth Factor Receptor (EGFR) which is overexpressed in PCa tissues at advanced stages. Therefore, enriched EGFR in blood can be used as a noninvasive biomarker that reflects the state of the disease in PCa patients [70]. Khan et al. reported that survivin, an oncoprotein associated with chemoresistance, is overexpressed in patient-derived exosomes and acts as a diagnostic and prognostic marker of PCa [71]. More interestingly, exosomes isolated from serum of African American men with PCa are a wealthy source of biomarkers for early detection and monitoring PCa patients [72]. In addition, exosomes isolated from urine can be utilized as a sentinel to monitor PCa stages and reduce the number of unnecessary biopsies [73].

A previous report indicated that miR-1290 and miR-375 in plasma have a potential role in the prediction of CRPC patients [74]. Exosomal miR-34a predicts docetaxel-treatment failure; that is, this miR contributes to the sensitivity of PCa cells to docetaxel through the downregulation of Bcl-2 [75]. Like miRNAs, exosomal proteins such as annexin A2, calsyntenin 1, fatty acid synthesis, filamin C, folate hydrolase-1, and growth differentiation factor 15 (GDF15) are specific for PCa diagnosis [67]. In addition, exosomal survivin is considered a promising biomarker for the early detection of PCa [71]. Different studies have elucidated the role of exosomes in chemo-resistance [76,77], radio-resistance [78], and immune-resistance [79,80]. It has been suggested that to override the negative effect of exosomes in cancer treatment, exosomes could be depleted from the blood of cancer patients using a hemodialysis-like technique [81]. In addition, exosomes could offer advantages in cancer therapy, such as being used as vehicles for drug delivery. The targeting of cancer cells by specific antibodies or ligands of highly expressed membrane receptors raised against cancer-associated exosomes constitutes an additional example of their therapeutic applications. Exosomes can be used to shuttle miRNAs, siRNA, and anticancer drugs directly to the targeted tumor cells. For example, exosomes loaded with siRNA are able to specifically target neuronal cells in murine brain [82]. In another study, exosomes were successfully delivered Let-7a miRNA to breast cancer cells in a xenograft mouse model [83].

Defining the role of exosomes in cancer biology has gained significant momentum in recent years, although relatively few studies have focused on the potential role of exosomal cargos in the recruitment of PCa neighboring cells to support tumor expansion and metastasis in the “seeds, soil and fertilizer” model. In this review, we have summarized the previously conducted studies on cancer-associated exosomes to better understand the role of bioactive molecules transported by exosomes as they bridge the crosstalk between tumor cells and their TME. The ultimate goal is to offer alternative or complementary therapeutic strategies that account for the TME as a vital component of the tumor architecture.

### 3.2. Singing Together: PCa Cell-Cell Interaction via Exosomal Signals

Cancer cells communicate with each other via exosomal cargo, which may include signaling complexes, receptors, functional proteins, or genetic information that regulates the signal networks involved in cancer growth and aggressiveness [84,85]. The two-way talk between sister tumor cells ominously affects the advancement of cancer and determines the response of cells to treatment options [86]. Several studies show that exosomes released from cancer cells promote tumor cell proliferation. Soekmadji et al. reported that the increase in androgen-deprived LNCaP cell proliferation was correlated with the release of CD9-positive exosomes [87]. Another study showed that LNCaP- and DU-145-derived exosomes can induce cell proliferation, epithelial-mesenchymal transition (EMT), migration, and IL-8 secretion while decreasing apoptosis in PCa cells [88,89,90]. In addition, hypoxic PCa cell-derived exosomes convey information that can be directly involved in the invasiveness and motility of dormant PCa cells [91]. Moreover, the exosomes released by different PCa cells confer a number of functional proteins to recipient cells, which do not originally express these proteins. It has also been shown that integrins are shuttled by exosomes isolated from PC-3 (αυβ6 and αυβ3 integrins) and CWR22 (αυβ3 integrin) cells to DU-145 and C4-2B cells that normally do not express integrin, which subsequently induces their progression and invasion [92,93]. More interestingly, exosomes are directly involved in the transport of chemotherapy resistance to other cells, as evidenced by a study conducted by Shedden et al. In this study, drug-resistant, cancer cell-associated exosomes were loaded with multidrug-resistant MDR-1 or p-gp proteins, which triggered the transfer of chemo-resistance to the recipient cells [76]. The exosomes released from the DU-145 and 22RV1 cells, which were resistant to doxorubicin, were not only able to transfer chemo-resistance to parental DU-145 and 22RV1 cells but also to LNCaP cells [94]. Furthermore, the impaired chemo-sensitivity to therapeutic agents is a reversible phenomenon in PCa. For example, when DU-145 cells with resistance to camptothecin or paclitaxel were treated with exosomes isolated from normal prostatic epithelial cells, including PrEC or the immortal cell line RWPE-1, their sensitivity was partially recovered [95,96]. Confirming this phenomenon, DU-145 cells with resistance to paclitaxel partially lost their resistance when treated with human mesenchymal stem cell (hMSC)-derived exosomes [96]. The different mechanisms by which exosomes opt to modify the genetic traits of PCa cells are outlined in Figure 1.

Regarding exosomal cargo, exosome-associated miRNAs are emerging as a novel regulator for many cellular functions, including cell metabolism. For example, miR-126, which is an angiogenesis inducer, regulates cancer metabolism via its downstream target insulin receptor substrate-1 (IRS1) [97,98]. IRS1 is not only a metabolic and growth-promoting protein via regulation of the insulin receptor (IR) and insulin-like growth factor I receptor (IGF-IR), but also contributes to neoplastic transformation [99]. Typically, endothelial cells (ECs) express and release miR-126 at a high level in the TME [100,101]. Notably, the level of exosome-associated miR-126 is elevated during glucose deprivation and oxidative stress [98,102,103]. In addition, the ectopic expression of miR-126 increases cellular glycolysis and mitochondrial dysfunction, downregulates Akt and FOXO1 signaling pathways, and upregulates gluconeogenesis and oxidative stress defense in malignant mesothelioma cells [97,98,104]. Cancer-associated exosomes containing miR-122 inhibit glycolytic metabolism by surrounding cells in the pre-metastatic niche via the regulation of pyruvate kinase muscle isozyme M2 (PKM2) and then allowing glucose uptake by growing cancer cells [105]. Further studies are warranted to validate the impact of PCa-derived exosomes in the metabolic crosstalk between cancer and stromal cells, which contributes not only to cell survival when oxygen and nutrients are deprived, but also to cancer progression and aggressiveness.

### 3.3. Effect of Exosomal Cargo on the Metabolic Reprogramming of Stromal Cells in the TME

Altered cell metabolism is a hallmark of cancer progression. According to the Warburg effect, cancer cells have a high tendency to ferment glucose, which is a process associated with lactate production, even in the presence of oxygen, that subsequently results in a reduced pH in the TME and, therefore, leads to cancer progression and aggressiveness [106]. There is increasing concern regarding the role of tumor-derived exosomes in the modulation of stromal cell metabolism. Inside the TME, metabolic coordination between cancer cells and stromal cells, such as cancer-associated fibroblasts (CAFs), tumor-associated macrophages (TAM), bone-marrow-derived cells (BMDCs), and tumor endothelial cells (TECs), plays a significant role in cancer cell survival and growth [107]. Most often, the adaptation of stromal cells to glycolysis supports cancer growth via the exchange of exosomes that provide the cancer cells with metabolic intermediates such as lactate, pyruvate, ketones, and glutamine, which can be used by cancer cells for the biosynthesis of macromolecules [107]. In this venue, fibroblasts, which represent one-third of stromal cells, are key players in cancer development [108,109,110]. Normal, dormant fibroblasts negatively affect the growth of tumor cells by maintaining epithelial homeostasis and proliferative latency [111,112]. Tumorigenic cells tend to induce pseudo-hypoxia by inducing HIFα, reactive oxygen species (ROS), and different oncogenic signals within the TME. These events are followed by the recruitment of normal-associated fibroblasts (NAFs) and their reprogramming into cancer-associated fibroblasts (CAFs), which are also known as myofibroblasts [113]. CAFs give-and-take both signaling molecules and metabolic fuels with the cancer cells, either by secreting individual molecules such as lactate or via the transfer of exosomal cargo. This process regulates the metabolic activities in the neighboring cancer cells and forcing them to respire and overcome their energy depletion [108,109,114]. Cancer-derived exosomes induce the Warburg effect and increase the rate of glycolysis, followed by lactate production in stromal cells [115]. One research team reported that CAF-derived exosomes promote a metabolic shift in PCa cells through the inhibition of mitochondrial oxidative phosphorylation. This caused an increase in glycolysis and the production of a variety of metabolites such as lactate, acetate, amino acids, tricarboxylic acid cycle intermediates, and lipids [116].

### 3.4. Desmoplastic Response of Stromal Cells to the PCa-Derived Exosomal Proteome

Essentially, the consequence of the mutual interaction between cancer cells and stromal cells on the TME favors cell survival, proliferation, angiogenesis, resistance to therapy, immune avoidance, and metastasis. Ample published data regarding the biology of exosomes have demonstrated that exosomes released by cancer cells reprogram their TME [117]. Undeniably, tumor-stromal cell crosstalk is always associated with tumor aggressiveness and the morphological transformation of stromal cells [118]. Proteomic studies of cancer-associated exosomes by advanced mass spectrometry have indicated that many cytoplasmic, membranous, Golgi apparatus, and endoplasmic reticulum proteins can be encapsulated in exosomal vesicles [119,120,121,122]. Cancer-associated exosomes opt to transport specific transmembrane proteins, including integrins and tetraspanins such as CD9, CD63, CD81, and CD82 that are specifically recognized by target cells, which can explain the high rate of exosomal uptake by cancer-adjacent stromal cells [117,123]. Interestingly, cancer-associated exosomes transport many functional proteins such as endosomal network proteins, which include (i) membrane transport and fusion proteins (GTPases, annexins, Rab proteins, and flotillin), (ii) heat-shock proteins (HSPs60, 70, and 90), (iii) multivesicular bodies (MVBs) biogenesis proteins (Alix and TSG101), (iv) cytoskeletal proteins (actin, tubulin, syntenin, and moesin), (v) lipoproteins and phospholipases, (vi) metabolic enzymes, and (vii) signal transduction proteins and major histocompatibility complement antigens [124,125,126,127].

In general, PCa-associated exosomes procured from clinical samples reveal cargo that contains cancer-related proteins such as CD9, CD81, and TSG101, Annexin A2, Fatty Acid Synthase (FASN), and prostate-specific membrane antigen (PSMA: a PCa-specific biomarker) [128,129]. PCa cell-derived exosomes transport Ras superfamily of GTPases Rab1a, Rab1b, and Rab11a to the TME, which then contribute to tumorigenic reprogramming and the recruitment of adipose-derived stem cells (ASCs), thereby supporting PCa cell growth and clonal expansion [130]. Emerging evidence shows that PCa-associated exosomes shuttle inactive TGFβ1, which induces a pro-tumorigenic phenotype differentiation of normal fibroblasts to CAFs via Mothers against Decapentaplegic Homolog 1 (SMAD)-dependent and -independent signaling pathways [131,132,133,134,135]. PCa-associated exosomes can also trigger the differentiation of bone marrow-mesenchymal stem cells (BM-MSC) to CAF cells, causing cells to become more active by producing high levels of VEGFA, HGF, and matrix metallopeptidase (MMP), which are associated with tumor growth [136]. Moreover, the treatment of cells with PCa-associated exosomal TGFβ1 increases the aggressive phenotype in CAFs compared to cells treated with the soluble form [132]. PCa-associated exosomes under hypoxic conditions contain nearly three times as much protein (CD63, CD81, HSP90, HSP70, Annexin II, TGF-β2, TNF1α, IL6, TSG101, Akt, ILK1, and β-catenin) as exosomes in normoxic conditions, which ultimately promotes CAF formation [91]. Furthermore, PCa-associated exosomes deliver the integrin αvβ3 to the TME where this integrin triggers the activation of Src phosphorylation and enhances the expression of pro-inflammatory S100 in stromal cells [93,137]. Src-family kinases are usually expressed in the prostatic epithelium, with their expression increasing during PCa initiation and progression, thus it is suggested that this kinase family is linked to normal cell transformation [138]. Many studies have demonstrated the vital role of c-Src tyrosine kinase, insulin-like growth factor 1 receptor (IGF-1R), and focal adhesion kinase (FAK) in PCa development and angiogenesis [139,140]. Remarkably, PCa-associated exosomes are enriched by c-Src, IGF-1R, and FAK proteins [141]. Aggressive PCa-associated exosomes contain urokinase-type plasminogen activator (uPA) [142], a stimulator for a protease plasminogen that is linked to vascular structure remodeling [143]. Moreover, PCa-associated exosomes promote the escape of tumors from immune surveillance by compromising the cytotoxic function of lymphocytes and reduce NKG2D receptor expression on natural killer and CD8+ T cells [144]. Cancer-associated exosomes not only shuttle immune regulatory molecules such as FasL, TGF-β, galectin-9, and HSP72, which help cancer cells escape the immune system, but also trigger the Fas/FasL pathway to induce CD8+ T cells toward the apoptotic pathway [145,146]. One recent study revealed that oncosomes (100–400 nm in diameter) in addition to exosomes can modify the prostate stroma phenotype [147]. Minciacchi et al. reported that the uptake of large oncosomes by prostate fibroblasts induces an αSMA-positive CAF phenotype that is independent of other CAFs markers such as MMP1, thrombospondin-1 (TSP-1), and TGFβ1, proving that oncosomes induce a distinct CAF phenotype [148], as outlined in Table 1 and Figure 2.

### 3.5. Exosomal Lipids as Messengers in PCA-Stromal Cell Crosstalk

Lipid metabolism is overwhelmingly disturbed in cancer cells compared to normal cells. Exosomes reflect the composition of different lipid types in cancer cells. Lipids have been implicated in different aspects of exosome biogenesis and various functions. The exosomal membrane is a lipid bilayer enriched in lipid raft-like domains. These domains act as platforms for lipid raft-associated proteins [161], diglycerides, sphingolipids, and glycerophospholipids including phosphatidylcholine (PC), phosphatidylserine (PS), phosphatidylethanolamines (PE), and phosphatidylinositol (PI) [162]. Of note, exosomes gain their rigidity from their lipid composition, which gives them stability in extracellular fluids, and also facilitates the process of internalization by recipient cells [162,163]. Although similarities exist in the orientations of membranous molecules between the exosomal membrane and their parental cell membranes, nonetheless the distribution of PS in the exosomal membrane (enriched in the outer leaflet) is in contrast to the distribution of these molecules in the membrane of live parental cells (inner leaflet) [162]. It has been shown that exosomal prostaglandins (PGE2) are involved in immunosuppression and can trigger tumor growth [164]. Interestingly, the transfer of phosphatidylcholine transporter ABCA3 by exosomes has been indicated in B lymphoma treatment that is resistant to immunotherapy with rituximab [79]. Moreover, exosomal lipids induce apoptosis in SOJ-6 human pancreatic cancer cells via inhibition of the Notch-1 pathway [165]. Exosomal lipids also increase drug resistance in human pancreatic cancer MiaPaCa-2 cells through the C-X-C motif chemokine receptor 4 (CXCR4)/stromal cell derived factor (SDF)-1α signaling pathway [166]. Finally, exosomal lipids can be considered as biomarkers in PCa [167]. However, further studies are warranted to define the lipid composition of PCa-associated exosomes and whether the lipid composition becomes altered at different tumor stages, which could affect the biological functions of exosomes in stromal cells

### 3.6. The Role of Exosomes in Shuttling Bioactive Materials between PCA and Stromal Cells

Exosome-mediated plane transfer of genetic and epigenetic materials comprising multiple RNA species such as messenger RNA (mRNA), microRNAs (miRNAs), long noncoding RNAs (lncRNAs), ribosomal RNA (rRNA), piwi RNA (piRNAs), small-nuclear RNA (snRNA), small-nucleolar RNA (snoRNA), and transfer RNA as well as genomic and mitochondrial DNA among cancer cells and the stromal cells are significant pathways for maintaining a favorable environment for cancer growth [66,168,169]. miRNAs are evolutionarily known as short, noncoding, single-stranded RNAs of 22 nucleotides in length with partial homology to sequences in their target mRNA. miRNAs regulate the posttranscriptional level of gene expression via the formation of a silencing complex (RISC), which allows annealing of miRNAs to the 3′UTR target genes and consequently represses protein expression. The second choice of posttranscriptional regulation occurs through mRNA destabilization [170,171,172]. Emerging evidence has shown that exosomal miRNAs regulate many biological functions in the TME such as the desmoplastic response of stromal cells and the proliferation, apoptosis, and invasion of tumor cells [173]. One study revealed that PCa cell-associated exosomes deliver onco-miRNAs including miR-125b, miR-130b, and miR-155 and onco-mRNAs such as H-ras and K-RAS transcripts, which have a desmoplastic effect on ASCs, thereby supporting tumor progression and colonization [130]. Moreover, miR-155 can suppress the expression of its target protein 53-induced nuclear protein 1 (TP53INP1) to promote CAF-like phenotypes in fibroblasts [174].

The cargo of PCa cell-derived exosomes not only induces the expression of prostate-specific genes in bone marrow cells (BMCs) but also in normal human cells [175]. More specifically, PCa-derived exosomes transport miR-100, miR-21, and miR-139, which upregulate the expression of receptor activator of nuclear factor kappa-Β ligand (RANKL) and Metalloproteinases in CAFs and promote PCa growth and metastasis [176]. Metastasis-initiating cells (MICs) are distinct circulating tumor cells (CTCs) with a strong ability to grow, survive, and colonize PCa cells in distant metastatic organs [177]. MIC-derived exosomes can modify the TME to stimulate the oncogenic transformation of normal epithelial and stromal cells, which stimulate phenotype transformation and promote PCa epithelial-mesenchymal transition (EMT) via the activation of RANKL, FOXM1, and c-Myc [177,178,179]. In addition, metastatic PCa patient-derived exosomes express a high level of miR-141 and miR-21, which regulate osteoclastogenesis and osteoblastogenesis [180,181]. One group of researchers showed that PC-3-derived exosomes regulate osteoclastogenesis and osteoblast proliferation and promote bone metastasis [182]. Another research group revealed that EVs transfer miR-409, miR-379, and miR-154, which have a vital role in embryogenesis and pluripotent stem cell formation, and favor PCa carcinogenesis and metastasis by activating tumor-stroma interactions [177].

Long non-coding RNAs (lncRNA), measuring more than 200 nucleotides in length, have a potential role in many physiological and pathophysiological processes [183]. In a comparative study using the lncRNA array, PCa cell line-associated exosomes overexpress 26 lncRNAs compared to normal epithelial cells [184]. These exosomal lncRNAs are significantly enriched in target motifs for the miRs, which are found in the same exosomes, suggesting that the sorting of exosomes is a highly organized process that allows for the selection of specific miRs and lncRNAs to be uploaded in their vesicles [184]. It has been reported that exosome-associated lncRNA MYU induces adjacent PCa cell proliferation and migration by competitively binding to miR-184 and therefore upregulates c-Myc [185]. Moreover, urine-derived exosomes collected from PCa patients are enriched in lncRNA-p21, which is associated with the malignancy of PCa [186]. In addition to the reported studies summarized in Table 2 and Figure 2, the role of PCa-derived exosomal lncRNA cargo in the recruitment of stromal cells that are still in the growth phase remains to be understood, such that further studies are needed to fully address this phenomenon.

### 3.7. Two-Way Crosstalk between Stromal Cells and PCA Cell-Associated Exosomal Cargo

It is noteworthy, that PCa-associated CAFs promote PCa progression, metastasis and development of resistance to therapy [191,192], as summarized in Figure 3.

Several mechanisms are suggested to explain the possible role of CAFs in promoting PCa progression. The first mechanism anticipates the role of CAFs in supporting PCa progression by increasing extracellular matrix (ECM) deposition and turnover in CAFs, which are accompanied by the production of cytokines such as TGF-β [193]. The second mechanism arises due to the overexpression of CAF growth and angiogenic factors, such as growth/differentiation factor 15 (GDF15) [194]. The third mechanism is a significant increase in the release of soluble factors and insoluble ECMs by CAFs, which promote the neoplastic transformation of normal cells to PCa-like cells [195]. In addition, CAF-derived exosomes transfer miRNAs into neighboring epithelial cells, which caused tumor growth in a PCa-mouse model through the EMT pathway [196,197]. CAF-derived exosomes transfer miRNAs and proteins, including miR-21, miR-409, and CD81 to adjacent epithelial cells and promote cell proliferation, invasion, and chemo-resistance and alter metabolic pathways in PCa cells [145,196,197]. Interestingly, miR-21 represses the expression of its target’s apoptotic peptidase activating factor 1 (APAF1) and programmed cell death 4 (PDCD4) to inhibit apoptosis and increase paclitaxel resistance in cancer cells [198]. In the same context, osteoblast-derived exosomes can promote PC-3 cell proliferation [182]. In connection to the previous studies, the subcutaneous co-injection of ARCaPE cells (which naturally lack miR-409) in combination with miR-409-expressing stromal fibroblast cells promoted ARCaPE cell proliferation when compared to control cells, suggesting that exosomal miR-409 can be transferred from stromal cells to PCa cells [196]. A similar study using exosomes isolated from menstrual stem cells (MenSCs) to treat PC-3 cells induced the inhibition of VEGF on mRNA and protein levels, decreased ROS production, and the secretion of HIFα and, therefore, suppressed PCa cell growth [199].

## 4. The Immune-Modulatory Role of Exosomes

The immune system is one of the main barriers to develop cancer cells. Hence, complex communication between immune cells and growing tumor cells is crucial for cancer initiation, progression, and metastasis. Delineating the role of exosomes in regulating immune cells during and after cancer development is needed to develop advanced exosome-based vaccines and immune therapy. Neoplastic modulations of the immune cells include up-regulation of precise genes and proteins and subsequent escape from immune cell recognition and killing [200,201,202]. Inside TME, cancer cells are able to remodel cytotoxic T lymphocytes (CTLs) and natural killer (NK) cells to facilitate tumor progression [203,204]. It was reported that cancer-associated exosomes interfere with the development of CD14^+^ monocytes into mature dendritic cells (DCs) [205]. More interestingly, the activated myeloid-derived suppressor cells (MDSCs) decreased NK and CD4^+^/CD8^+^ lymphocytes and reduced their cytotoxic effect through the interaction between exosomal HSP72, TLR-2, and MyD88 and immune cells [81]. Cancer-associated exosomes not only shuttle immune regulatory molecules such as PD-L1, TRAIL, TGF-β, IL-10, FasL, galectin-9, HSP72, and PGE2 to exert an immunosuppressive effect, but they also to trigger the Fas/FasL pathway and induce apoptosis in CD8^+^ T cells [145,146,206,207,208,209,210]. Cancer-associated exosomes can also alter the behavior of macrophages, which regulate host immunity. This promotes tumor progression by releasing cytokines, inducing tissue remodeling, and promoting angiogenesis and metastasis [211,212,213]. Moreover, PCa-associated exosomes promote tumor escape from immune surveillance by compromising the cytotoxic function of lymphocytes and reducing NKG2D receptor expression in NK and CD8^+^ T cells [144]. On the contrary, circulating NK and CD8^+^ T cells in CRPC patients decrease surface killer cell lectin-like receptor K1 (KLRK1) expression. Therefore, treating healthy lymphocytes with CRPC serum-derived exosomes decreases the expression of KLRK1 in effector lymphocytes [144]. In the same scenario, cancer-associated exosomes shuttle particular miRNAs such as miR-24-3p, miR-891a, miR-106a-5p, miR-20a-5p, and miR-1908, which compromise T-cell function in nasopharyngeal cancer [214,215]. Furthermore, pancreatic cancer-associated exosomes shuttle miR-212-3p to inhibit regulatory factor X-associated protein (RFXAP), a regulatory transcription factor for the MHC-II gene, and initiate immune tolerance in dendritic cells (DCs) [216]. In addition, an in vivo study conducted on pancreatic cancer revealed that exosomal miR-203 downregulates the expression of TRL4, affects TNF-α and IL-1 production, and deteriorates DC development and Th1 differentiation [217]. In another study, engineered exosomes with TNF-related apoptosis-inducing ligand (TRAIL) rewired the apoptotic pathway in vitro and in a preclinical mouse model [218].

## 5. Targeting the Tumor Microenvironment

TMEs share common features such as hypoxia, oxidative stress and acidosis, uncontrolled growth, a resistance to apoptosis, a metabolic shift toward anaerobic glycolysis and the remodeling of ECM-associated cells [219]. In addition to other biological factors, exosomes secreted by cancer and TME cells modulate stromal cells residing in the TME [220,221]. Understanding such changes in the TME during tumor progression will yield additional advantages in the development of novel therapeutic strategies to tackle cancer progression and metastasis compared to the currently available options, for example, by targeting the ECM by using angiotensin II, TGF-β, and heparanase inhibitor roneparstat (SST0001) [222,223,224,225,226], affecting hypoxia and acidosis by targeting hypoxia-induced factor-1 (HIF-1) using several compounds and therapies such as Topotecan [227,228,229], and targeting endothelial cells and pericytes to avoid neovascularization using several antiangiogenic drugs such as bevacizumab (Avastin) [230,231]. In addition, targeting the recruitment of tumor-associated-macrophages (TMAs) in the TME [232], activating the anti-tumor activity of immune cells [233], and targeting CAFs [230] might add new therapeutic benefits in the treatment of tumors in advanced stages. In the same context, targeting cancer-associated exosomes may open new venues in the treatment of aggressive forms of PCa. Toward this direction, Bastos and colleagues discussed different means of targeting exosomes during the process of exosomal biogenesis, release, and uptake [234]. Tumor-associated exosomes can be targeted during the process of biogenesis and released by inhibiting ceramides using the GW4869 inhibitor [235] and Rab-associated proteins [236]. Manumycin A blocks biogenesis and secretion of exosomes by inhibiting the activity of ERK1/2 and expression of heterogeneous nuclear ribonucleoprotein H1 (hnRNPH1) in CRPC cells [237]. In addition, blocking the mechanisms of exosome internalization is another approach for targeting cancer-associated cells in the TME. For example, methyl-β-cyclodextrin (MCβD) inhibits exosome uptake in a number of cancer cells by disrupting the lipid rafts of exosomes [238]. Cytochalasin D, heparin, dynasore, and nystatin are other examples of exosome uptake inhibitors [234]. However, mechanisms of targeting exosomes are very complicated because some of these pathways are involved in most cellular activities. Therefore, further studies are warranted to shed light on the biology of exosomes to find more specific targets.

## 6. Conclusions and Future Directions

The advancement of science and technology opens new venues for understanding new mechanisms underlying cancer initiation, progression, and metastasis and, therefore, offers new treatment options for improving the survival of cancer patients. One important scientific direction in recent years has been to shed light on the various roles of exosomes in cancer biology, in addition to their clinical applications. The mechanisms by which tumor cells send and receive molecular signals from the TME is a growing area of research. A significant quantity of research has indicated that exosomal cargo is a powerful communication means in cell-cell interactions, and is an integral part of the crosstalk between tumor and stromal cells in the TME. Given that exosomes contribute to cancer progression through the transfer of different types of exosomal cargo to their target cells, these bioactive molecules stimulate multiple oncogenic pathways, which remodel normal, cancer, and stromal cells inside the TME. Elucidating the underlying mechanisms of cancer aggressiveness by shuttling exosomal cargo between PCa and stromal cells in the TME increases the viability of using exosomes as promising therapeutic agents. In addition, these vesicles can serve as vectors for many anticancer agents, a) because of their biological nature and cellular origin, b) because of lipophilic nature that makes them able to easily cross membranes such as the blood-brain barrier, c) because they are not filtered by the glomerulus, which gives exosomes long circulation lifetimes, and d) because exosomes can carry different types of bioactive molecules. Exosomal cargo can be considered as potential diagnostic and prognostic markers for different types of cancers, which has created excitement in the scientific community because this fact paves the way for using exosomal cargo as reliable tumor markers. Another application is to use exosomes to unwire the connection between cancer cells and the TMEs, with the potential to reduce PCa metastasis and the development of drug resistance. Precise information regarding how exosomes selectively load their cargo, independent of their mother cells, and then release and internalize these bioactive molecules into target cells will reshape our treatment modalities for cancer patients. Although a growing number of institutes and agencies share a wealth of biological data on public domains, more collaborative studies and access to these exosomal data remain in demand. The assembly of multidisciplinary research teams is highly encouraged to understand the biology of the TME and clinical utilities of exosomes in the context of cancer disease and its effective treatment.

## Figures and Tables

**Figure 1 cells-09-00564-f001:**
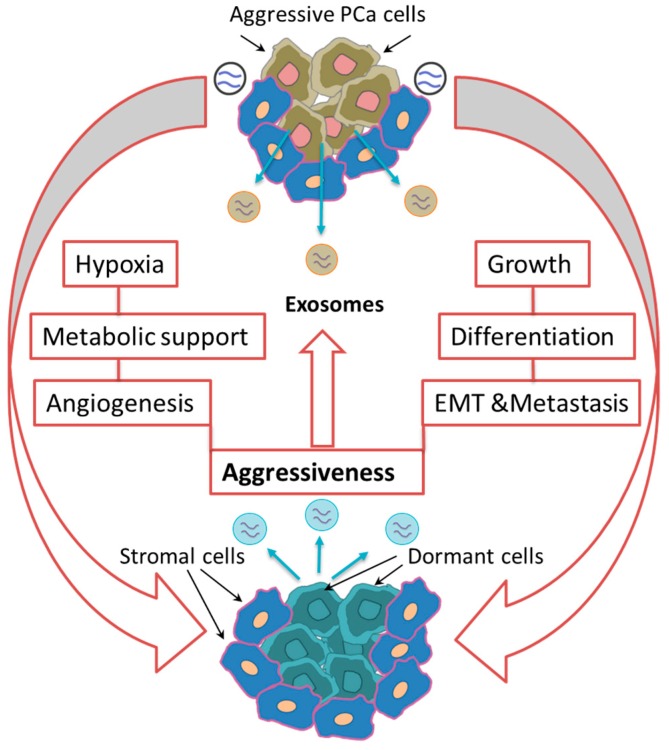
Birds in a nest: Role of exosomes in intra-prostate cancer (PCa) communications. PCa cells communicate with each other by transfer of exosomal cargo proteins and nucleic acids within the tumor microenvironment. In heterogenic tumor cells and in premetastatic niche, shedding of exosomes from vicious PCa cells reactivate dormant PCa cells to gain new aggressive traits through inducing of cell growth, differentiation, epithelial-mesenchymal transition (EMT), metabolic adaptation under hypoxic conditions, and angiogenesis.

**Figure 2 cells-09-00564-f002:**
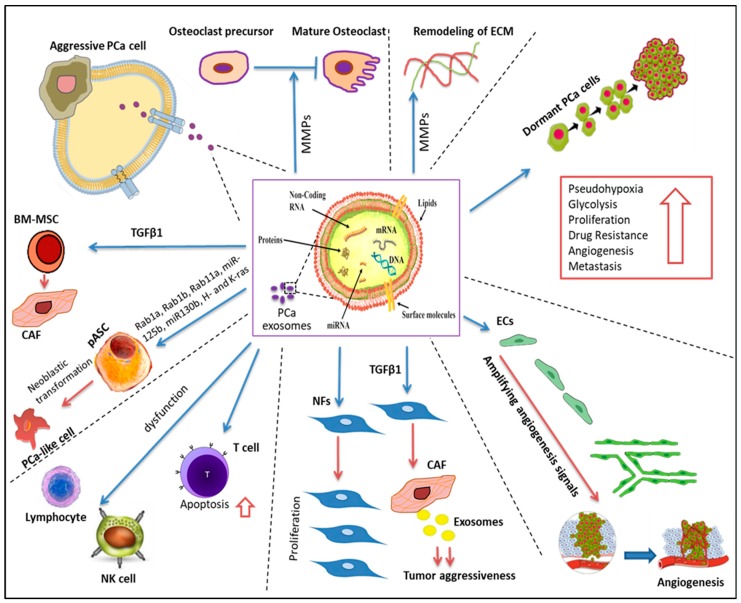
The potential roles of prostate cancer (PCa)-associated exosomes in modulating different cells in the TME. Exosomes-associated cargo shuttles bioactive molecules to and from PCa cells to activate stromal cells in the TME for gaining new genetic traits and promoting PCa progression and metastasis. Fibroblast cells (FB), cancer associate fibroblast (CAF), endothelial cells (ECs), patient adipose derived stem cells (pASC), T cell, natural killer cells (NK), lymphocytes, bone marrow-mesenchymal stem cells (BM-MSC), osteoclast cells, extracellular matrix (ECM). Part of this figure was prepared by the aid of Mind the GRAPH program available on https://mindthegraph.com/.

**Figure 3 cells-09-00564-f003:**
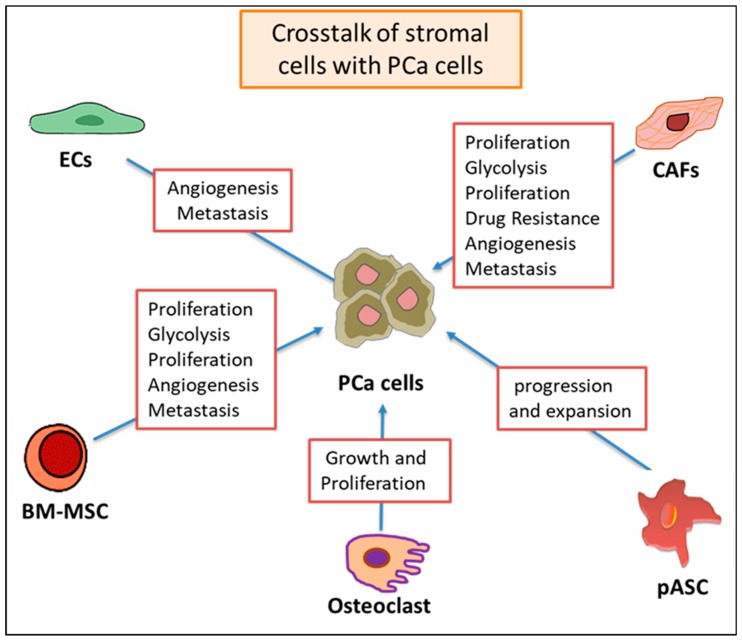
Effect of stromal cell-derived exosomes on PCa progression and metastasis. Stromal cells release exosomes fully loaded with cargo macromolecules to promote cell proliferation, hypoxic and adaptive metabolic pathways, angiogenesis and metastasis of PCa cells. Cancer-associate fibroblasts (CAFs), endothelial cells (ECs), patient adipose-derived stem cells (pASCs), bone marrow-mesenchymal stem cells (BM-MSC) and osteoclast cells. Part of this figure was prepared by the aid of Mind the GRAPH program.

**Table 1 cells-09-00564-t001:** Exosomes-associated cargo proteins and their functional relevance in Tumor microenvironment (TME).

Protein(s)	Biological Function	Reference(s)
AHNAK	Cancer-associated exosomes increase motility of fibroblasts	[149]
ANXA2, CLSTN1, FASN, FLNC, FOLH1, and GDF15	Correlated with PCa malignancy	[67]
CD9	Increase the proliferation and chemo-resistance of PCa cells	[87,150]
B7-H3	Immune checkpoint regulator	[151]
CD63, CD81, HSP90, HSP70, TNF1α, IL-6, MMP2, MMP9, Annexin II, TSG101, Akt, ILK1, and β-catenin	Increase stemness, metastasis and CAFs formation	[91]
c-Src, IGF-1R and FAK	PCa development and angiogenesis	[139,140,141]
Ets-1	Induce osteoclast differentiation	[152]
FasL, TGF-β, galectin-9 and HSP72	Evade immune responses	[145,146]
Galectin-1	Promote angiogenesis	[153]
Integrin αvβ3	Pro-inflammatory effect on stromal cells	[93,137]
Integrin β4, vinculin and P-gp	Associated with taxane and docetaxel resistance	[154,155]
MMP14	Promote PCa cell growth	[156]
PD-L1	Immune checkpoint regulator	[157]
Rab1a, Rab1b and Rab11a	Neoplastic transformation of pASCs	[130]
TGF-β	Differentiation of fibroblast to CAFs	[131,132,133,134,135]
TIMP-1	Associated with PCa aggressiveness	[158]
Trop-2, vimentin, N-cadherin and Integrin αvβ3	Induce PCa cell invasion	[92,136,159,160]
uPA	Vascularization remodeling of PCa microenvironment	[142,143]
VEGFA, HGF and MMP	Angiogenesis, EMT and tumor growth	[136]

**Table 2 cells-09-00564-t002:** PCa-associated exosomes transfer nucleic acids in their cargo, which have significant functions in TME.

Nucleic Acid(s)	Biological Function	Reference(s)
miR-125b, miR130b, miR-155	Neoplastic transformation of PCa patients’ adipose stem cells	[130]
miR-125b	Downregulate AKT1 expression and induce PCa proliferation	[187]
miR-100, miR-21 and miR-139	PCa growth and metastasis	[176]
miR-141, miR-21, miRNA-375	Affect osteoclastogenesis and osteoblastogenesis and help PCa cells to overcome androgen deprivation in long distant metastasis	[180,181,188,189]
miR-409, miR-379 and miR-154	Support PCa carcinogenesis and metastasis	[177]
miR-485-3p	Associated with fludarabine resistance in PCa cells	[190]
H-ras, K-ras (mRNA)	Neoplastic transformation of PC patients’ adipose stem cells.	[130]
LincRNA-p21 (lncRNA)	Associated with PC malignancy	[186]
lncRNA MY (lncRNA)	Promote adjacent cell proliferation and migration	[185]

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
