# Peer review of "Exosomes are the Driving Force in Preparing the Soil for the Metastatic Seeds: Lessons from the Prostate Cancer"

_cells, 2020, doi:10.3390/cells9030564_

Round 1
Reviewer 1 Report
In this manuscript the authors summarize the role of exosomes on the tumor microenvironment (TME). They describe exosomal composition, signaling in prostate cancer cells, metabolic reprograming of stromal cells, lipid and mRNA cargo. There is an absence of discussion of how exosomes modulate the immune system. The review does a good job summarizing exosomal crosstalk and the potential role in drug resistance, angiogenesis and metastasis. However, the reviewers have left out discussing the potential role in suppressing/modulating the anti-tumor immune response.
Specific Comments
1) Font size change at line 110-125
2) Figure 1 image has border around cells. Please ensure this is an original drawing and not clip art that may violate copyright. Similar image issues arise in figure 2 as well.
3) The authors should consider including a section for modulation of the immune system. With a discussion of exosomes inhibiting T cell and NK cell function (PMID:25268476) and inducing T-cell tolerance (PMID:31269655). Please include immune modulators in Table1, such as immune checkpoint regulators B7-H3 (PMID:18829542) in prostate and PD-L1 in other cancers (PMID:30089911).
4) The authors should explain the composition of exosomes more thoroughly in section 3.1 and describe how exosome could signal through membrane bound proteins or release of cargo (miRNA, protein). The authors should also describe the difference (size, secretion process...) between EVs and microvesicles.
5) Please elaborate on the different means of targeting exosomes during the process of exosomal biogenesis, release, and uptake in section 4. “Targeting the tumor microenvironment”
Author Response
Response to the Reviewers comments
Thank you very much for reviewing our Review article and providing us with invaluable comments. It is always valuable for authors to receive a good review and constructive comments. In our case, we appreciate the Editor and Reviewers’ time and efforts to promote out manuscript onto a publishable stage. Based on these comments and suggestions, we have made very careful modification in the previous version of our manuscript entitled “Exosomes are the driving force in preparing the soil for the metastatic seeds: Lessons from prostate cancer.” The details of the modifications are highlighted in yellow in the revised manuscript and the itemized corrections are listed and discussed point-by-point for your kind consideration.
Reviewer #1
- General Comments
In this manuscript the authors summarize the role of exosomes on the tumor microenvironment (TME). They describe exosomal composition, signaling in prostate cancer cells, metabolic reprograming of stromal cells, lipid and mRNA cargo. There is an absence of discussion of how exosomes modulate the immune system. The review does a good job summarizing exosomal crosstalk and the potential role in drug resistance, angiogenesis and metastasis. However, the reviewers have left out discussing the potential role in suppressing/modulating the anti-tumor immune response.
Our response: We thank the Reviewer for these comments and we complied with the Review’s request and we added a new section “The immune-modulatory role of exosomes” in which we discussed the role of exosomes in suppressing the immune cells during cancer development (please see the highlighted section 4 in page 13).
“4. The immune-modulatory role of exosomes
The immune system is one of the main barriers to develop cancer cells. Hence, a complex communication between the immune cells and growing tumor cells is crucial for cancer initiation, progression and metastasis. Delineating the role of exosomes in regulating immune cells during and after cancer development is needed for development of advanced exosomes-based vaccines and immune therapy. Neoplastic modulations of the immune cells include up-regulation of precise genes and proteins and subsequently escape from immune cell recognition and killing [1-3]. Inside TME, cancer cells are able to remodel cytotoxic T lymphocytes (CTL) and natural killer (NK) cells to facilitate tumor progression [4, 5]. It was reported that cancer-associated exosomes interfere with the development of CD14+ monocytes into mature dendritic cells (DC) [6]. More interestingly, the activated myeloid-derived suppressor cells (MDSCs) decrease NK and CD4+/CD8+ lymphocytes and reduce their cytotoxic effect through the interaction between exosomal HSP72, TLR-2 and MyD88 and immune cells [7]. Not only cancer-associated exosomes shuttle immune regulatory molecules such as PD-L1, TRAIL, TGF-β, IL-10, FasL, galectin-9, HSP72 and PGE2 to exert an immunosuppressive effect, but also to trigger Fas/FasL pathway and induce apoptosis in CD8+ T cells [8-14]. Cancer-associated exosomes can also alter the behavior of macrophages, which regulate host immunity. This promotes tumor progression by releasing cytokines, inducing tissue remodeling, and promoting angiogenesis and metastasis [15-17]. Moreover, PCa-associated exosomes promote tumor escape from immune surveillance by compromising the cytotoxic function of lymphocytes and reduce NKG2D receptor expression in NK and CD8+ T cell [18]. In contrary, circulating NK and CD8+ T cells in CRPC patients decrease surface NKG2D expression. Therefore, treating healthy lymphocytes with CRPC serum-derived exosomes decrease the expression of NKG2D in effector lymphocytes [18]. In the same scenario, cancer-associated exosomes shuttle particular miRNAs such as miR-24-3p, miR-891a, miR-106a-5p, miR-20a-5p, and miR-1908 which compromise T-cell function in nasopharyngeal cancer [19, 20]. Furthermore, pancreatic cancer-associated exosomes shuttle miR-212-3p to inhibit the regulatory factor X-associated protein (RFXAP), a regulatory transcription factor for MHC-II gene, and initiate immune tolerance in dendritic cells, DCs [21]. In addition, an in vivo study conducted on pancreatic cancer revealed that exosomal miR-203 downregulates the expression of TRL4, affect TNF-α and IL-1 production and deteriorate DCs development and Th1 differentiation [22]. In another study, engineered exosomes with TNF-Related Apoptosis-Inducing Ligand (TRAIL) rewire the apoptotic pathway in in vitro and in preclinical mouse model [23].”
- Specific Comments
1) Font size change at line 110-125
Our response: We apologize for this change and we corrected it.
2) Figure 1 image has border around cells. Please ensure this is an original drawing and not clip art that may violate copyright. Similar image issues arise in figure 2 as well.
Our response: We apologize for this fault and we fixed it in the figures. Also, we acknowledged the website we used to draw some components of the figures as following:
“Part of this figure was prepared by the aid of Mind the GRAPH program available on https://mindthegraph.com/
3) The authors should consider including a section for modulation of the immune system. With a discussion of exosomes inhibiting T cell and NK cell function (PMID:25268476) and inducing T-cell tolerance (PMID:31269655). Please include immune modulators in Table1, such as immune checkpoint regulators B7-H3 (PMID:18829542) in prostate and PD-L1 in other cancers (PMID:30089911).
Our response: We included these references under section 4 “The immune-modulatory role of exosomes on page 13 and in Table 1, page 17. We thank the Reviewer for bringing up these very important reports to our attention.
4) The authors should explain the composition of exosomes more thoroughly in section 3.1 and describe how exosome could signal through membrane bound proteins or release of cargo (miRNA, protein). The authors should also describe the difference (size, secretion process...) between EVs and microvesicles.
Our response: We thank the Reviewer for bringing up this excellent comment although the intention of this review was to focus on the role of exosomes in promoting aggressive forms of prostate cancer. We included concise information on exosomes structure and signals (please see the highlighted yellow new information incorporated to section 3.1, page 4)
“EVs are small double-membrane bodies released by normal and abnormal cells and classified into three main types based on the size of vesicles. The typical size of EVs ranges from 100nm to 1µm, exosomes from 30 to120nm and apoptotic bodies from 500nm to 2µm in diameter [24]. Exosomes are intraluminal vesicles that are derived from multivesicular bodies through a process of endosome ripening, in which the vesicles either fuse with lysosomes to degrade their cargo or fuse with the plasma membrane to release exosomes into the extracellular matrix [25-27]. The number and composition of exosomes depend on the physiological cellular activities in which exosomes are involved in it. Exosomes can be characterized by a set of exosomes markers expressed on the outer membrane of these vesicles or enclosed in their cargos. The most common protein contents of exosomes are tetraspanins (CD63, CD9, CD81 & CD82), ESCRT-I associated protein (TSG101), lysosomes-associated membrane glycoproteins (LAMP-1 & 2B), MVB-associate protein (Alix-1), heat shock proteins (HSP60, 70 & 90), adhesion molecules (CD54 & CD11b), major histocompatibility molecules (MHC-1 & II), Ras-related proteins Rabs and membrane-binding proteins (annexins) [28]. In addition to proteins, exosomes carry in their cargos RNAs (microRNAs, mRNAs and lncRNAs), DNAs and lipids, which are previously reported [29].”
5) Please elaborate on the different means of targeting exosomes during the process of exosomal biogenesis, release, and uptake in section 4. “Targeting the tumor microenvironment”
Our response: We thank the Reviewer for providing this excellent comment. We elaborated on the different means of targeting exosomes during the process of exosomal biogenesis, release, and uptake in section 4 (highlighted yellow in Page 14).
“Toward this direction, Bastos and colleagues discussed different means of targeting exosomes during the process of exosomal biogenesis, release, and uptake [30]. Tumor-associated exosomes can be targeted during the process of biogenesis and release by inhibiting ceramides using GW4869 inhibitor [31] and Rab-associated proteins [32]. Manumycin A blocks biogenesis and secretion of exosomes by inhibiting the activity of ERK1/2 and expression of hnRNP H1 in CRPC cells [33]. In addition, blocking the mechanisms of exosomes internalization is another approach for targeting cancer-associated cells in the TME. For example, methyl-β-cyclodextrin (MCβD) inhibits exosomes uptake in a number of cancer cells by disrupting the lipid rafts of exosomes [34]. Cytochalasin D, heparin, dynasore and nystatin are other examples of exosomes uptake inhibitors [30]. However, mechanisms of targeting exosomes are very complicated because some of these pathways are involved in most of cellular activities. Therefore, further studies are warranted to shed lights on biology of exosomes to find more specific targets.”
Reviewer 2 Report
The manuscript of Saber et al is an interesting overview regarding the role of exosomes in cancer progression , metastasis..with a specific focus in PCa However, there are some important points that the Authors should clarify.
1) Fig 1 is not clear, aggressive PCa cells and dormant PCa cells are sister cells in the same tumor area? The authors should better clarify/describe these 2 specific cell subtypes
2) In general, it seems that the vast majority of available data regards cell lines and in vitro studies; very few data are available regarding exosomes isolated from PCa patients ; the authors should discuss with more details these few papers..
3) line 412-412: the authors should describe some of these "different means"
4) Please check the font size, it' is different between the various paragraphs of the text
Author Response
Response to the Reviewers comments
Thank you very much for reviewing our Review article and providing us with invaluable comments. It is always valuable for authors to receive a good review and constructive comments. In our case, we appreciate the Editor and Reviewers’ time and efforts to promote out manuscript onto a publishable stage. Based on these comments and suggestions, we have made very careful modification in the previous version of our manuscript entitled “Exosomes are the driving force in preparing the soil for the metastatic seeds: Lessons from prostate cancer.” The details of the modifications are highlighted in yellow in the revised manuscript and the itemized corrections are listed and discussed point-by-point for your kind consideration.
Reviewer #2
- General Comments
The manuscript of Saber et al is an interesting overview regarding the role of exosomes in cancer progression , metastasis..with a specific focus in PCa However, there are some important points that the Authors should clarify.
Our response: We appreciate the very positive comments received from the respected Reviewer.
- Specific Comments
1) Fig 1 is not clear, aggressive PCa cells and dormant PCa cells are sister cells in the same tumor area? The authors should better clarify/describe these 2 specific cell subtypes
Our response: We thank the Reviewer for this comment and we included more description for the two sister cells (See the updated Fig. 1).
Figure 1. Birds in a nest: Role of exosomes in intra-PCa communications. PCa cells communicate with each other by transfer of exosomal cargo proteins and nucleic acids within the tumor microenvironment. In heterogenic tumor cells and in premetastatic niche, shedding of exosomes from vicious PCa cells reactivate dormant PCa cells to gain new aggressive traits through inducing of cell growth, differentiation, epithelial-mesenchymal transition (EMT), metabolic adaptation under hypoxic conditions, and angiogenesis.
2) In general, it seems that the vast majority of available data regards cell lines and in vitro studies; very few data are available regarding exosomes isolated from PCa patients; the authors should discuss with more details these few papers.
Our response: We appreciate the Reviewer’s comment. We complied with the Reviewer comment and we included more information for exosomes isolated from PCa patients (highlighted yellow in page 5).
“It was reported that PCa patient-derived exosomes shuttle EGFR, which is overexpressed in PCa tissues at advanced stages. Therefore, enriched EGFR in blood can be used as noninvasive biomarker that reflects the state of the disease in PCa patients [35]. Khan et al. reported that survivin, an oncoprotein associated with chemoresistance, is overexpressed in patient-derived exosomes and acts as a diagnostic and prognostic marker of PCa [36]. More interestingly, exosomes isolated from serum of African American men with PCa are a wealthy source of biomarkers for early detection and monitoring PCa patients [37]. In addition, exosomes isolated from urine can be utilized as a sentinel to monitor PCa stages and reduce the number of unnecessary biopsies [38].”
3) line 412-412: the authors should describe some of these "different means"
Our response: We complied with the Reviewer comment and we included more information for different means of exosomes communications (highlighted yellow in page 5).
“In the same context, exosomes can communicate between two different cells through; 1) transfer of bioactive molecules in their cargo to activate/suppress signaling pathways in target cells, 2) receptors shuttling between donor and recipient cells to alter cellular activities, 3) transfer of fully functional proteins to perform specific functions in target cells, and 4) providing new genetic information to recipient cells to gain new traits [39].”
4) Please check the font size, it' is different between the various paragraphs of the text
Our response: We complied with the Reviewer’s request and adjusted the font size.
We thank the Reviewers again for their invaluable comments,
- The Authors